# Mode of HIV acquisition among adolescents living with HIV in resource-limited settings: A data-driven approach from South Africa

Eda He[1], Janke Tolmay[1]*, Siyanai Zhou[1,2], Wylene Saal[1], Elona Toska[1,3,4]

1 Centre for Social Science Research, University of Cape Town, Cape Town, South Africa, 2 School of Public Health, Department of Health Sciences, University of Cape Town, Cape Town, South Africa, 3 Department of Sociology, University of Cape Town, Cape Town, South Africa, 4 Department of Social Policy and Intervention, University of Oxford, Oxford, United Kingdom

* Janke.Tolmay@uct.ac.za

**Data Availability Statement:** As the study has not yet concluded, an anonymised dataset is not yet available to be shared publicly. Due to the sensitive nature of the information in the dataset, restrictions

## Abstract

Adolescents living with HIV (ALHIV) face unique treatment and care challenges which may differ by how they acquired HIV, whether vertically (in-utero, perinatal or postnatal exposure during breastfeeding) or sexually (sexual exposure). Distinguishing and documenting the mode of HIV acquisition (MOHA) is crucial to further research on the different needs and outcomes for ALHIV and to tailor HIV services to their needs. Age-based cut-offs have been used to attribute MOHA but have not been validated. We analysed data from a three-wave cohort of n = 1107 ALHIV part of a longitudinal study in South Africa. Age-based MOHA was allocated using age at ART initiation, validated against a logic-tree model based on literature-hypothesised factors: self-reported HIV, sexual, and family history. After testing six ART initiation age cut-offs (10 to 15 years old), we determined the optimal MOHA cut-off age by calculating the sensitivity and specificity for each cut-off, measured against the final logic-tree allocation. Following validation using this longitudinal study, the methodology is extended to 214 additional third-wave participants—adolescent girls and young women living with HIV who became mothers before the age of 20. Finally, descriptive statistics of the final allocations are presented. Among the 1,063 (96.0%) cohort study participants classified, 68.7% acquired HIV vertically, following validation. ART initiation before cut-off age 10 had the highest sensitivity (58.9%) but cut-off age 12 had the largest area under the curve (AUC) (0.712). Among the additional young mothers living with HIV, 95.3% were estimated to have acquired it sexually, following the same algorithm. For this group, while cut-off ages 10 to 12 had the highest sensitivity (92.2%), age 14 had the highest AUC (0.703). ART initiation before 10 years old is strongly associated with vertical HIV acquisition. Therefore, a cut-off age of 10 would remain the recommendation in LMIC regions with similar epidemiology as South Africa for determining MOHA in research and clinic settings.

on sharing still apply, as imposed by the ethical committee. More information about the study and ethics procedures are available online https://www.heybaby.org.za/contact, as well as relevant contact details. Data will be made available upon request for non-profit use following data access protocols laid out in our website https://www.heybaby.org.za. Based on specific requests, a de-identified and anonymized version of the data will be shared.

**Funding:** This study received funding from: the International AIDS Society through the CIPHER grant (155-Hod; 2018/625-TOS); Claude Leon Foundation [F08 559/C]; funded by Evidence for HIV Prevention in Southern Africa (EHPSA), a UK aid programme managed by Mott MacDonald; Janssen Pharmaceutica N.V., part of the Janssen Pharmaceutical Companies of Johnson & Johnson; the Nuffield Foundation, but the views expressed are those of the authors and not necessarily the Foundation; Oxford University Clarendon-Green Templeton College Scholarship; the Regional Inter-Agency Task Team for Children Affected by AIDS - Eastern and Southern Africa (RIATT-ESA); the Economic and Social Research Council [IAA-MT13-003]; the John Fell Fund [103/757; 161/033]; the Leverhulme Trust [PLP-2014-095]; the University of Oxford's ESRC Impact Acceleration Account [1602-KEA-189; K1311-KEA-004]; UNICEF Eastern and Southern Africa Office (UNICEF-ESARO); UKRI GCRF Accelerating Achievement for Africa's Adolescents (Accelerate) Hub (Grant Ref: ES/S008101/1); Oak Foundation/GCRF "Accelerating Violence Prevention in Africa" [OFIL-20-057]. Additionally, ET received funding from the European Research Council (ERC) under the European Union's Horizon 2020 research and innovation programme (grant agreement No 771468) and the Fogarty International Center, National Institute on Mental Health, National Institutes of Health under Award Number K43TW011434. The funders had no role in study design, data collection and analysis, decision to publish, or preparation of the manuscript.

**Competing interests:** The authors have declared that no competing interests exist.

# Background

Mode of HIV acquisition (MOHA) can be classified as either vertical or sexual. The Centres for Disease Control and Prevention define vertical acquisition as the transmission of the HIV virus from parent to child either during gestation, birth, or through breastfeeding, while horizontal (sexual) acquisition is defined as transmission through blood, semen, pre-seminal fluid, rectal fluid, or vaginal fluid [1]. Recent studies have found differences in disease phase, clinical and social support needs for HIV management, and ART adherence by MOHA among adolescents living with HIV (ALHIV) [2–5]. For example, experiences of transitioning in healthcare services vary by MOHA, as vertically infected adolescents who initiate care with paediatric services may find entering adult care difficult due to orphanhood and drastic differences between paediatric and adult services, while adolescents with sexually acquired HIV initiate HIV care through adult services that are unfamiliar and do not acknowledge their unique needs, while simultaneously facing the issues of stigma and disclosure that come with a new diagnosis, especially during adolescence or pregnancy [6, 7]. Similarly, the clinical care needed may differ as vertically infected patients are more likely to have transitioned beyond first-line treatment compared to sexually infected patients who have been infected or on treatment for a shorter length of time [8]. This emphasizes the need for differentiated service delivery (DSD), a client-centred approach that aims to tailor HIV programmes to best meet the diverse needs of people living with HIV while alleviating the burden on health care systems [9]. Distinguishing and documenting MOHA is critical to further research on the different needs and outcomes for ALHIV, to adapt services to better support ALHIV and interrupt the cycle of HIV transmission. Care must be taken, however, to avoid potentially stigmatising language when referring to MOHA of adolescents, as terms such as 'behaviourally' or 'perinatally' infected fail to convey the nuance of their clinical circumstance or their unique individual needs [10].

In 2018, UNAIDS estimated that there were 1.3 million ALHIV globally, 88% of them in sub-Saharan Africa [11]. Although the risk of early mortality is increased for vertically infected children who are not initiated on ARV, research has shown that approximately 30% of vertically infected adolescents remain asymptomatic for 10 to 16 years [12–14]. These 'slow progressors' provide justification for delayed HIV testing and delayed antiretroviral therapy (ART) initiation among some vertically infected adolescents. Moreover, rates of new infections among 15-19-year-old adolescents persist, resulting in a growing cohort of sexually infected adolescents in sub-Saharan Africa. The adolescent HIV incidence rate was 29 cases per 100,000 in 2018 and projections indicate 1.8 million adolescents will be newly infected between 2018 and 2030 [11, 15]. While the number of adolescents with sexually acquired HIV is increasing, healthcare services are not adjusting to these changing demographics [15, 16].

In low-resource settings, collecting information regarding MOHA is difficult, with patients attending multiple clinics, and the common use of paper-based medical records which are prone to data entry errors, misspecification, and misclassification. Thus, it is critical to develop a low-cost and effective method for determining MOHA among adolescent populations using self-reported information to ensure the appropriate care and research accuracy.

To determine HIV MOHA in low resource settings, most cohort studies have used the age cut-off of <10 years old at the time of ART initiation for vertical acquisition and ≥10 years old for sexual acquisition [17]. More recent analyses consider shifting the cut-off age to 15 years old at the time of ART initiation, which may be especially pertinent as the age of enrolment into HIV care is, on average, higher in the sub-Saharan Africa region due to delayed rollout of large-scale ART [5, 18, 19]. Using medical records combined with self-reported personal history data from a representative ALHIV cohort in the Eastern Cape province, South Africa, we aim to determine MOHA of ALHIV using age at ART initiation validated against self-reported

ART experiences, family history, and sexual practices. We test the sensitivity and specificity of six different ART initiation cut-off ages, ranging from the previously recommended age 10 to the newly recommended age 15, for optimal MOHA determination.

## Methods

### Sample and study setting

This analysis first uses three rounds of data from the Mzantsi Wakho longitudinal study conducted with the ethical approval of the Universities of Cape Town (CSSR 2013/4; 2019/01) and Oxford (SSD/CUREC2/12-21), as well as the Eastern Cape Provincial Departments of Health and Education. Adolescents who had ever initiated ART in the public health sector in the Buffalo City municipality of the Eastern Cape, South Africa, were community-traced to be invited to face-to-face interviews. Baseline reached 90.1% of eligible adolescents, aged 10 to 19 years old, with follow-up rates of 93.6% and 91.5% in two consecutive follow-up rounds. Following an explanation of the study and consent protocol, voluntary written informed consent for observations, interviews, and accessing medical records was obtained from the adolescent participant and a caregiver for adolescents <18 years old, by means of electronic signatures on data collection tablet devices. Participant confidentiality was preserved, except in instances of reported or observed abuse, in which case they were referred to appropriate support services. Medical records were collected by a trained clinic team capturing patient files at local clinics and hospitals within the municipality for 88.1% of adolescents [20]. Additional information on the study procedures is described elsewhere [21].

The analysis is extended to a further 214 newly-recruited ALHIV—young mothers who are interviewed at the third wave of Mzantsi Wakho between March 2018 and July 2019. These form part of the new HEY BABY (Helping Empower Youth Brought up in Adversity and their Babies and Young Children) study data, collected using a broad and comprehensive systematic sampling strategy which included district health facilities, maternity units, secondary schools, and social worker or community referrals. Voluntary written informed consent was also obtained from these additional participants. Based on their self-reported data, the 214 young mothers were initiated on ART, but none had information on their MOHA and were therefore included in further analyses and allocation. Since we only had data for these participants at one wave, the MOHA algorithm was adapted considering that these participants were already mothers and older at the time of the survey.

All analyses were conducted using Stata 16 and R version 3.6.2.

### Determination of Mode of HIV Acquisition (MOHA)

To determine MOHA, we first allocated participants based on an ART initiation age cut-off: 10, 11, 12, 13, 14, and 15 years old. A person who initiated at a self-declared age younger than the cut-off was determined to have acquired HIV vertically while initiating at the cut-off age or older was classified as sexual acquisition (in this paper we use "vertically acquired" to refer to adolescents living with HIV infections resulting from parent-to-child transmission, and "sexually acquired" for adolescents living with HIV infections resulting from unprotected sexual intercourse). Next, a logic tree was created to validate the age cut-off allocations, using questions about the participant's sexual, family, and ART history. Using the logic tree allocations as a reference, we calculated the sensitivity, specificity, positive and negative predictive values, and area under the curve for each of the five age cut-offs using the initial age cut-off determinations. These calculated measures were compared across age groups to decide on an optimal cut-off age for MOHA determination. The measures are generally used to determine which

'test' (in this case age cut-off) provides a most accurate estimation of a 'gold standard' (the logic tree determination using personal history data).

Sensitivity is defined as the proportion of participants who have a certain condition, and give positive test results, while specificity refers to the proportion of participants who do not have a certain condition, and then test negative. The positive predictive value stipulates the proportion of those who tested positive who are truly positive, while the negative predictive value refers to those who tested negative and are true negatives [22, 23]. In the current study, we define true positives as participants with sexual HIV acquisition, while true negatives are those with vertical acquisition.

### Logic tree decision-making algorithm

The logic tree was constructed using factors typically associated with either vertical or sexual HIV acquisition, based on previous literature. The branches are ordered with the most highly correlated factors first (see Fig 1). The gateway branch used to decide whether a person would undergo the vertical logic tree or sexual logic tree was whether the adolescent reported having had sexual activity and/or experienced sexual assault at any of the three data collection points, as non-vertically infected patients in this cohort were most likely to have acquired HIV via sexual routes. At the end of the logic tree branching, final MOHA determinations were confirmed based on the presence of corroborating and countering information.

A similar method was used for the additional participants, but since all these participants were mothers, the entire group initially entered the sexual infection logic tree (right arm) based on the gateway question of sexual activity (Fig 1). After following the logic tree branching, the tree-based MOHA determinations in this group were compared to the same age cut-off MOHA determinations as described above.

To ascertain the strength of the evidence that led to the MOHA determination in the logic tree, we calculated the frequencies of each branch at which the determination was made for

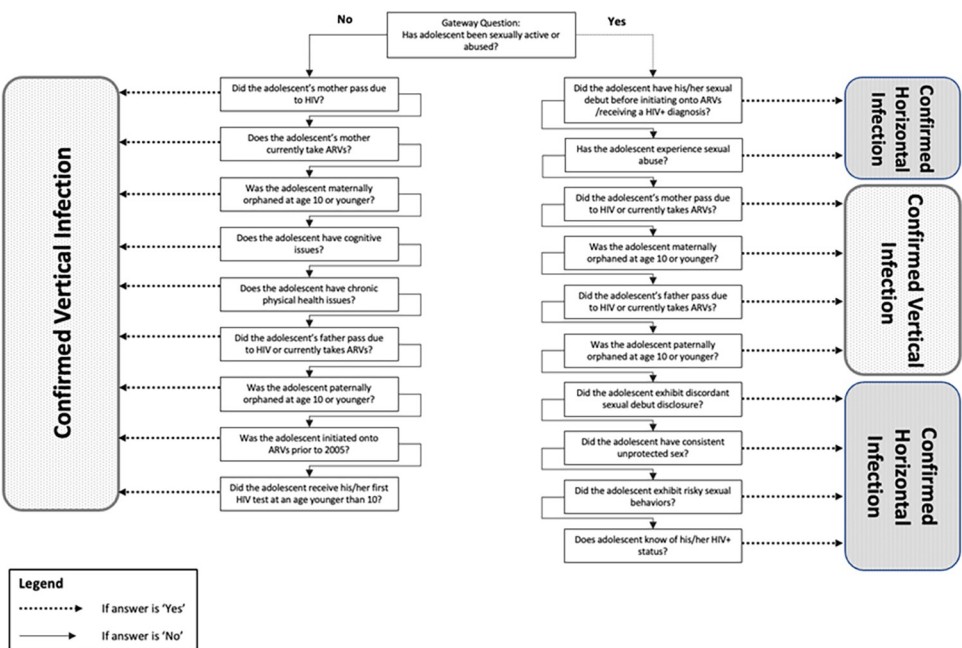

**Fig 1. Logic tree diagram.** Logic tree with each of the vertical acquisition side branches and sexual acquisition side branches to final MOHA determination.

both samples. A lower exit number indicates a determination was made at a higher branch, indicating stronger evidence. Although the participants' MOHA were determined once they reached an affirmatory factor, many adolescents had multiple factors aligned with their MOHA outcome, so we also calculated the average number of affirmatory factors for participants with sexually and vertically acquired HIV in each sample.

## Measures for factors included in the decision-making algorithm

**ART initiation age.** Age of ART initiation was determined from both medical records and self-reported data, using the youngest age across the three data collection waves. The medical records were searched and abstracted from clinics where adolescents accessed health care (adolescent's primary ART clinic). In the case of missing age of ART initiation in medical records, self-reported age of ART initiation was used [24].

**Sexual activity.** Adolescents who were sexually active prior to HIV-positive status disclosure or prior to ART initiation were confirmed as sexually infected. All three waves of data were used to determine adolescents' sexual debut age with sexual debut defined as vaginal or anal intercourse. Additionally, adolescents who had ever experienced penetrative sexual abuse were also confirmed as sexual acquisition given that the 'virgin cleansing' myth and child sexual abuse are relevant matters of concern in South Africa [25–28].

**Discordant disclosures of sexual debut.** Across all waves, the same questions about the age of first sexual intercourse (vaginal and anal) were asked. In Wave 1, the questions were asked by the research assistant. In Wave 2 and 3, the questions were asked using audio computer-assisted self-interview (ACASI) software to allow for confidentiality. Adolescents who either (1) did not report a sexual debut age in Wave 1 but reported a sexual debut age in Wave 2 or (2) reported a sexual debut age in Wave 1 that was 3 years or older than the sexual debut age they reported in the later waves, exhibited what was classified as discordant disclosure of sexual debut. These adolescents were already sexually active at Wave 1 but did not report any activity until Waves 2 and/or 3. Or, these participants reported an older age of sexual debut in Wave 1 but reported a different age of debut when they had the privacy to do so without revealing it to the research assistant in Waves 2 and/or 3 using ACASI. Adolescents with discordant disclosure of sexual debut were confirmed as having acquired HIV sexually, as previous research has shown stigmatized topics, such as sexual behaviour, are more likely to receive accurate responses via ACASI compared to face-to-face interviews [29, 30]. Sexual discordant disclosure suggests personal stigma as well as willingness to provide inaccurate information about other stigmatizing events, such as sexual HIV acquisition. Thus, discordant sexual disclosure confirmed sexual HIV acquisition. The additional 214 young mothers only had data at Wave 3, which means the consistency of the answers could not be determined over the three waves. Therefore, this branch was excluded from these participants' logic tree determination.

**Risky sexual behaviour.** Risky sexual behaviour factors were composed using data from all three waves. Past-year risky sexual behaviour was measured as having consistent unprotected sex, transactional sex, multiple partners, or partner(s) five or more years older than themselves. Poverty and food insecurity are drivers of risky sexual behaviour among adolescents and young people, who in turn are unable to negotiate safe sex [31, 32]. This increases the risk of sexual infection, so the presence of risky sexual behaviour was applied as confirmatory evidence of sexual HIV acquisition.

**Biological mother/father: HIV-infected & AIDS-related death.** Among adolescents undergoing the sexual portion of the logic tree who were not picked up by a previous decision node, maternal and paternal HIV infection was examined. Adolescents whose biological mother was taking ART, had HIV symptoms, or passed away due to HIV/AIDS in any of the

three waves were confirmed as vertically infected. The rate of prevention of mother-to-child transmission (PMTCT) during 1994–2005 was low, given that South Africa did not have a PMTCT program until after 2002 and vertical transmission was the more common mode of acquisition among children born at the time [33, 34]. Paternal HIV infection was used as a later node than maternal HIV infection, as the maternal genetic link is better established than the paternal link in this cohort [35].

**Maternal and paternal orphanhood.** After excluding participants who were maternally or paternally orphaned due to HIV/AIDS, orphanhood due to other reasons was checked. Participants who were orphaned and gave reasons such as, "I don't know", "illness", and "poison" for parental mortality, were assumed to have lost their parent due to HIV/AIDS. It is common in this generation of adolescents that the reasons for their parents' deaths remain undisclosed, and deaths related to HIV/AIDS are often attributed to witchcraft [36, 37]. To have stricter criteria, orphanhood reason and HIV/AIDS progression timeline were combined across the three waves to create the branching node.

**Time from seroconversion to AIDS death in adults pre-ART rollout by sex and rural residence.** A literature review was done to assess the average time from seroconversion to AIDS death amongst ART naïve males and females in low- and middle-income countries (LMICs). There are few studies in LMICs with recorded seroconversion dates, with most studies starting with HIV-infected cohorts, and different HIV-1 subtypes have been found to have different rates of disease progression [38–41].

Multiple studies in sub-Saharan Africa found subtype D to have a faster progression than any other subtype while all other subtypes have similar disease progression timelines [38, 40, 42]. South Africa's HIV epidemic is primarily subtype C, especially in those heterosexually infected [43–45]. One study in rural Masaka, Uganda that had primarily subtype A infections found the median time from seroconversion to AIDS death was 10.2 years, 9.4 years from seroconversion to AIDS, and 0.8 years from AIDS to death [46]. This figure was similar to other studies in LMICs [39, 40, 47–49]. Though studies on life expectancies of HIV-infected persons showed differences by sex, these studies were examining cohorts that had initiated ART [50]. Studies following HIV-infected, ART-naïve cohorts did not find significant differences in survival times by sex and rural residence [51–53].

Based on the 10.2 years from adult seroconversion to AIDS death progression, adolescents who indicated they were maternally or paternally orphaned (in any of the three waves) at age 10 and younger and provided "I don't know", "illness", or "poison" as reasons for parental orphanhood were confirmed as vertically infected because the evidence indicates the parents would have been HIV-infected at the time of their conception.

The age of the participant at either of their parents' deaths was only documented during Wave 1 for Mzantsi Wakho participants, which means that this information was not available for the additional participants recruited through the HEY BABY study. Therefore, these measures were excluded from this group's logic tree MOHA determination.

**Cognitive delay and poor physical health.** Compared to persons with sexually acquired HIV, participants with vertical acquisition experience significantly higher cognitive impairment due to infection during key neurodevelopmental phases [54–57]. Cognitive delay was measured by difficulty focusing and/or answering during the interview, parental report, and attendance at a special needs school. Similarly, poor physical health is linked to vertically infected persons due to side effects and long-term use of ART during key physical development ages [58, 59]. Poor physical health was measured by chronic illness and frequent infections including TB, using data from all three waves. Participants were confirmed as vertically infected if they had impacted cognitive development or poor chronic physical health.

**ART initiation year and HIV test age.** In South Africa, ART was initially only available in urban hospitals, where only doctors could prescribe it [60]. In the Eastern Cape, ART roll-out programs were started by non-profit organisations, such as Médecins Sans Frontières spearheaded by Hermann Reuters and the Keiskamma Trust, using private funds to initiate HIV-infected residents [61, 62]. It was not until 2005 that the government-funded rollout of ART and it would take another year before every South African district would have an HIV treatment facility [61, 63]. During this time, the criteria for initiating children on ART was either a CD4 count of <15–20% or Stage II or III based on the World Health Organization (WHO) Paediatric HIV & AIDS classification in 2004 [64, 65]. Thus, if an initially vertically determined adolescent was not determined by a previous decision node, ART initiation prior to 2005, calculated using the youngest ART initiation age across the three waves added to the birth year, was used to confirm vertical infection.

**'Slow progressors': HIV disease progression in ART naïve children and adolescents.** Adolescents in the vertically infected side of the logic tree who were ART naïve, but had their first HIV test when they were younger than 10 years old were confirmed as vertically infected slow progressors, given the disease progression of these adolescents matches those of 'slow progressors' or 'long-term non-progressors' described in the literature [12, 13].

## Results

Of the 1,107 ALHIV in the three-wave cohort, 57.0% were female, and the median age at Wave 1 was 13 years. The median age at ART initiation was 6 years old, and the median age of sexual debut (not including sexual violence) was 15 years old when considering the youngest age reported across all three waves (Table 1). Of the additional 214 new adolescent mothers included in Wave 3, the median age was 19 years, the median age at ART initiation was 18 years, and the median age at sexual debut was 16 years old. In both groups, approximately a quarter of adolescents lived in rural areas (Table 1).

## MOHA allocation and confirmation

For our cohort adolescents, using different ART initiation age cut-offs from age 10 to 15 we found that between 71.0% and 86.7% were vertically infected. Using the logic tree, 1,063 adolescents from the cohort were assigned a MOHA determination. Of the total cohort, 760 (68.7%) were determined by the logic tree as vertically infected, four of whom were classified

**Table 1. Descriptive statistics of cohort adolescents living with HIV (N = 1107), and young mothers added at Wave 3 (N = 214).**

|  | Adolescent cohort | | Additional young mothers | |
|---|---|---|---|---|
|  | N or Median | % or IQR | N or Median | % or IQR |
| *Sex* |  |  |  |  |
| Female | 631 | 57.0 | 214 | 100.0 |
| *Home Location* |  |  |  |  |
| Rural at Wave 1 | 281 | 26.3 |  |  |
| Rural at Wave 2 | 260 | 25.1 |  |  |
| Rural at Wave 3 | 244 | 24.2 | 50 | 23.4 |
| *Age* |  |  |  |  |
| Wave 1 | 13 | 5 |  |  |
| Wave 3 |  |  | 19 | 2 |
| *ART start age* | 5 | 9 | 18 | 3 |
| *Sexual debut age* | 15 | 3 | 16 | 2 |

**Table 2. Original HIV mode of acquisition allocation based on an age cut-off of 10 years and an adjusted allocation incorporating the logic tree decision-making algorithm for the adolescent cohort (non-mothers), adolescent cohort (mothers) and additional mothers.**

|  | HIV MOHA | Mzantsi Wakho cohort (non-mothers) (n = 958) | Mzantsi Wakho cohort (mothers) (n = 149) | Additional young mothers (n = 214) |
|---|---|---|---|---|
|  |  | N (%) | N (%) | N (%) |
| Original allocation* | Sexual | 199 (20.8) | 122 (81.9) | 194 (90.7) |
|  | Vertical | 759 (79.2) | 27 (18.1) | 20 (9.3) |
| Algorithm allocation** | Sexual | 182 (19.0) | 121 (81.2) | 204 (95.3) |
|  | Vertical | 732 (76.4) | 28 (18.8) | 10 (4.7) |
|  | N/A | 44 (4.6) | 0 (0.0) | 0 (0.0) |

*Age cut-off of 10 years old

**Logic-tree based allocation

as 'slow progressors'. Forty-four (4.0%) adolescents could not be validated and were hence coded as missing MOHA. When extracting the mothers from this group, the majority (81.2%) of the 149 cohort mothers were allocated to sexual acquisition, compared to only 19.0% of the 958 non-mothers (both boys and girls) (Table 2).

When measuring the strength of the evidence leading to MOHA determination in the logic tree, we found that most adolescents were confirmed by the first four decision branch factors in both the vertical and sexual groups (Table 3). Many adolescents had multiple factors congruent with their final MOHA determination made by the logic tree.

**Table 3. Frequency of adolescents' exit branch by mode of HIV acquisition determined by logic tree.**

|  | Cohort adolescents (N = 1063) | Additional young mothers (N = 214) |
|---|---|---|
| **Exit Branch** | N | N |
| *Sexually Acquired* | **303** | **204** |
| Exit1: Sex before HIV+ diagnosis/ initiating on ART | 122 | 132 |
| Exit2: History of sexual abuse | 96 | 1 |
| Exit3: Discordant sex disclosure | 6 |  |
| Exit4: Consistent unprotected sex | 56 | 63 |
| Exit5: Risky sex behaviour | 7 | 3 |
| Exit6: HIV+ status awareness | 16 | 5 |
| *Vertically Acquired* | **760** | **10** |
| Exit1*: Maternal HIV orphan | 357 | 10 |
| Exit2*: Mom on ART | 76 |  |
| Exit3*: Maternal orphan ≤10yo | 113 |  |
| Exit4: Cognitive issues | 109 |  |
| Exit5: Chronic physical disability | 13 |  |
| Exit6*: Paternal HIV orphan or dad on ART | 12 |  |
| Exit7*: Paternal orphan ≤10yo | 31 |  |
| Exit8: ART initiated before 2005 | 31 |  |
| Exit9: First HIV test age <10 | 15 |  |
| Slow Progressors | 3 |  |

*Includes adolescents who switched from the sexual side of the logic tree to the vertical side of the logic tree.

**Table 4. Average number of affirmative answers to each logic tree branch after adolescents are filtered into the vertical or sexual logic tree.**

| | Cohort adolescents (N = 1063) | | Additional young mothers (N = 214) | |
|---|---|---|---|---|
| | N (%) | Average affirmative answers (SD) | N (%) | Average affirmative answers (SD) |
| *Sexual Logic Tree* | **303 (100.0)** | **3.1 (1.1)** | **204 (100.0)** | **3.0 (0.78)** |
| Sex before HIV+ diagnosis/ initiating on ART | 122 (40.3) | | 132 (64.7) | |
| History of sexual abuse | 118 (38.9) | | 5 (2.5) | |
| Discordant sex disclosure | 29 (9.6) | | n/a | |
| Consistent unprotected sex | 231 (76.2) | | 168 (82.4) | |
| Risky sex behaviour | 168 (54.8) | | 97 (47.5) | |
| HIV+ status awareness | 276 (91.1) | | 201 (98.5) | |
| *Vertical Logic Tree**\* | **760 (100.0)** | **3.1 (1.5)** | **10 (100.0)** | **1.6 (0.84)** |
| Maternal HIV orphan | 290 (38.2) | | 10 (100.0) | |
| Mom on ART | 148 (19.5) | | 0 (0.0) | |
| Maternal orphan ≤10yo | 346 (45.5) | | n/a | |
| Cognitive issues | 387 (50.9) | | 0 (0.0) | |
| Chronic physical disability | 133 (17.5) | | 0 (0.0) | |
| Paternal HIV orphan or dad on ART | 136 (17.9) | | 2 (20.0) | |
| Paternal orphan ≤10yo | 212 (27.9) | | n/a | |
| Initiated before 2005 | 353 (46.4) | | 3 (30.0) | |
| First HIV test age <10 | 319 (42.0) | | 1 (10.0) | |

*Note some of the vertical factors are inherently exclusionary of each other (e.g. maternally orphaned and mother currently on ART).

Among the newly added young mothers, using age cut-offs of 10 to 12 showed that 90.7% of participants acquired HIV sexually, while cut-offs at 13, 14, and 15 years yielded 89.7%, 88.8%, and 83.6%, respectively. All of the 214 new participants were allocated to a MOHA using the adapted logic tree determination, with 204 (95.3%) categorised as sexually infected and 10 (4.7%) as vertically infected (see Table 2).

Using a similar method as for the cohort, when looking at the frequency of each confirmatory branch for the young mothers with sexually acquired HIV, Exit Branch 1 and Exit Branch 4 had the highest frequencies (Exit 3 was not included for the additional participants due to insufficient information for this branch). All the young mothers with vertical HIV acquisition were confirmed at Exit Branch 1 (see Table 3).

The average number of affirmatory factors among the cohort adolescents and additional young mothers are illustrated in Table 4.

## Determination of optimal cut-off age

Comparing the six ART initiation age cut-offs tested in the three-wave cohort, ART initiation at 10 years old had the highest sensitivity (correctly classifying sexually infected adolescents as sexual acquisition using the cut-off when compared to the logic tree determination) at 58.9%. Age 10 also had the smallest absolute difference between sensitivity and specificity (see Table 5).

In contrast to the cohort, sensitivity in the additional young mothers' group was high while specificity was low, indicating a high rate of false positives (incorrectly classifying vertical participants as sexual). The cut-off ages 10 to 12 had identical results and the highest sensitivity was 92.2%, while sensitivity and specificity were most closely aligned at age 15 (Table 6). Figs 2 and 3 indicate the receiver operating characteristic (ROC) curves for the 6 cut-off ages for each group, respectively.

**Table 5. Calculation of optimal cut-off age through validation against logic tree determination for the cohort adolescents.**

| Age cut off | Sensitivity | Specificity | Abs Difference* | PPV | NPV | AUC |
|---|---|---|---|---|---|---|
| 10 years | 0.589 | 0.815 | 0.226 | 0.562 | 0.831 | 0.702 |
| 11 years | 0.541 | 0.860 | 0.319 | 0.609 | 0.823 | 0.701 |
| 12 years | 0.522 | 0.902 | 0.380 | 0.683 | 0.824 | 0.712 |
| 13 years | 0.471 | 0.936 | 0.465 | 0.748 | 0.814 | 0.704 |
| 14 years | 0.427 | 0.955 | 0.528 | 0.793 | 0.805 | 0.691 |
| 15 years | 0.408 | 0.969 | 0.561 | 0.842 | 0.802 | 0.688 |

*Absolute difference between sensitivity and specificity

**Table 6. Calculation of optimal cut-off age through validation against logic tree determination for the additional young mothers.**

| Age cut off | Sensitivity | Specificity | Abs Difference* | PPV | NPV | AUC |
|---|---|---|---|---|---|---|
| 10 years | 0.922 | 0.400 | 0.522 | 0.969 | 0.200 | 0.661 |
| 11 years | 0.922 | 0.400 | 0.522 | 0.969 | 0.200 | 0.661 |
| 12 years | 0.922 | 0.400 | 0.522 | 0.969 | 0.200 | 0.661 |
| 13 years | 0.912 | 0.400 | 0.512 | 0.969 | 0.182 | 0.656 |
| 14 years | 0.907 | 0.500 | 0.407 | 0.974 | 0.208 | 0.703 |
| 15 years | 0.853 | 0.500 | 0.353 | 0.972 | 0.143 | 0.677 |

*Absolute difference between sensitivity and specificity

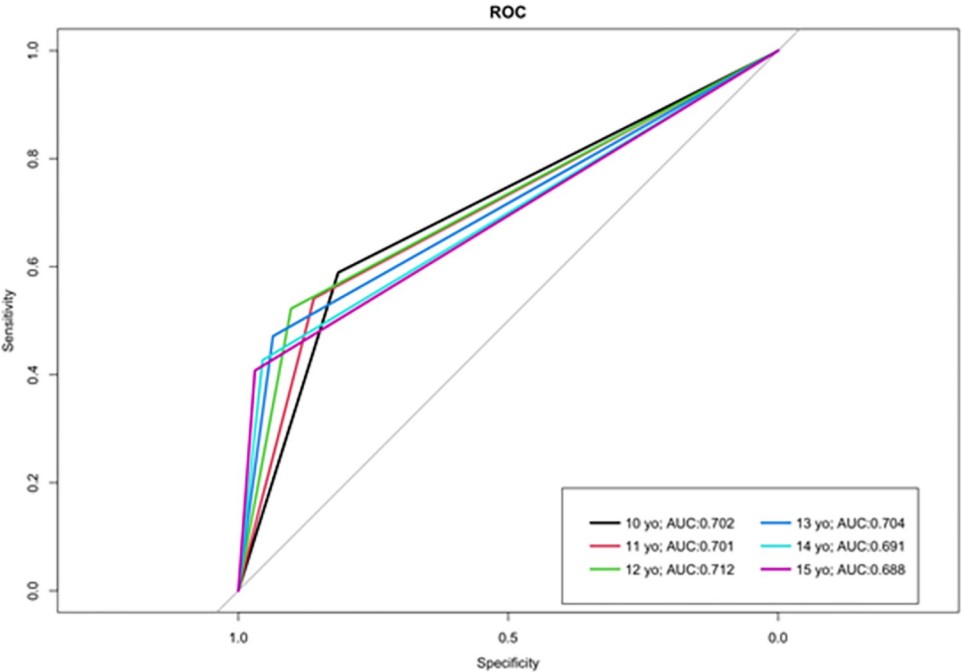

**Fig 2. Receiver Operating Characteristic (ROC) curves for the 6 MOHA age cut-offs in the cohort.** Receiver Operating Characteristic (ROC) Curves for the 6 MOHA age cut-offs in the cohort with their respective area under the curve (AUC) values.

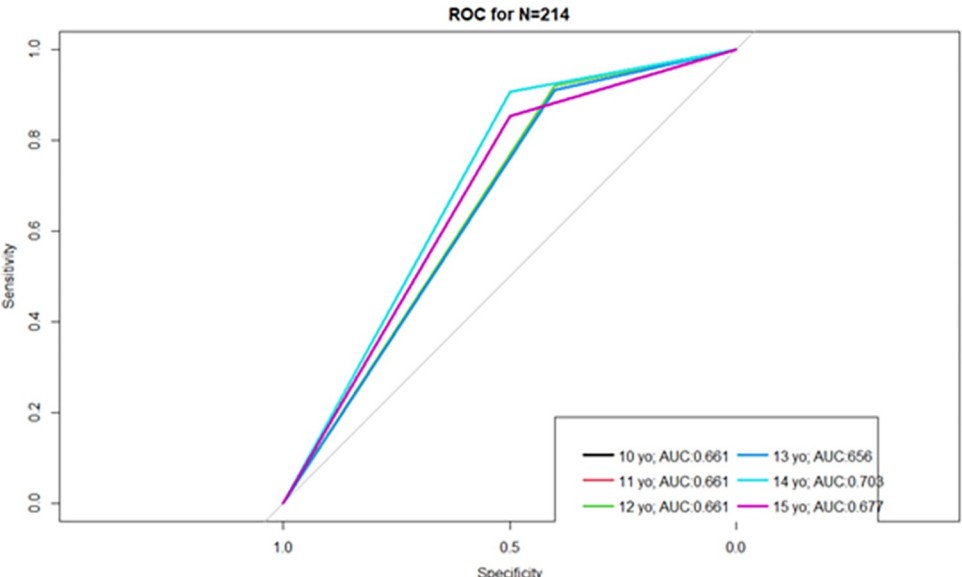

**Fig 3. Receiver Operating Characteristic (ROC) curves for the 6 MOHA age cut-offs in the additional young mothers.** Receiver Operating Characteristic (ROC) Curves for the 6 MOHA age cut-offs in the additional young mothers with their respective area under the curve (AUC) values.

## Discussion

Emerging evidence has found variations in health determinants and needs among adolescents living with HIV based on their mode of HIV acquisition. These determinants may include familiarity with the health system, available social support, and HIV status awareness [2, 7]. As ALHIV is a higher risk group for loss to follow-up and HIV-related mortality, research and clinical services should consider MOHA to understand differences in HIV pathology and needs to tailor services for better outcomes [19, 66–69]. Assessing MOHA can improve upon research on the effects of length of infection on long-term needs and outcomes of ALHIV in LMIC settings, as most of the current research on transition of care based on MOHA comes from high-income countries [7]. It can also help inform the provision of differentiated health services, such as support in transitioning from paediatric to adult care for adolescents with vertically acquired HIV, and sexual reproductive health and disclosure counselling for adolescents with sexual HIV acquisition [7, 24]. However, this is often not possible in settings with limited data systems, where ART rollout happened late or inconsistently, and where access to testing among HIV-exposed children is low. With the lack of means for accurate MOHA determination, an age cut-off method may be the only viable and pragmatic method. This analysis aimed to identify ART initiation ages that could be used in further research related to people living with HIV.

After incorporating self-reported personal history data, the logic tree found 68.7% of our cohort group to have acquired HIV vertically. This is in alignment with a recent study from five high-prevalence Southern African countries, which found that between 55%-73% of ALHIV were vertically infected [5]. Conversely, both the logic tree and age cut-offs found the majority of the additional adolescent mothers to have acquired HIV sexually (95.3% and 88.8%, respectively). When extracting the adolescent mothers from the Mzantsi Wakho cohort, we found the same trend, which is distinctly different from the non-mothers (boys and girls) in the same sample. This finding aligns with the multi-country results from nationally representative surveys which found higher rates of vertical MOHA among adolescent boys [5].

Vertically infected adolescents frequently present with physical and cognitive developmental challenges, including stunting, pubertal delay, and neurocognitive defects [70–73]. The impact of these delays on sexual activity and adolescent pregnancy may explain why most of the young mothers in both samples acquired HIV sexually, rather than vertically.

As seen by the results, sensitivity and specificity have a give and take relationship—an increase in sensitivity means lower specificity and vice versa. Adolescents who acquired HIV sexually are the hidden subpopulation overshadowed in the current healthcare structure. Thus, this paper recommends maximizing sensitivity. For the Mzantsi Wakho cohort, this indicates age 10 to be the optimal cut-off out of all the cut-off ages tested, which aligns with the original cut-off age proxy used by UNICEF. Age 10 also provides the optimal balance between sensitivity and specificity. In the group of adolescent mothers, while cut-off at age 15 had the smallest absolute difference between sensitivity and specificity, ages 10 to 12 had the highest sensitivity. Thus, age 10 remains an acceptable cut-off point for this group as well, since sensitivity is high, and the other indicators differ only marginally compared to ages 13 to 15. This would be most appropriate for populations with similar adolescent HIV epidemiology as South Africa.

Adolescents who acquired HIV sexually are more likely to need additional attention involving integrated education and clinical care due to their later entry into HIV care services and different entry points—most likely through antenatal care or sexual and reproductive healthcare services. The lack of differentiated and tailored health services could be contributing to the higher rates of loss to follow-up and virological failure among adolescents compared to children and adults [74]. In settings where some additional personal history may be acquired, the logic tree branches that had the strongest evidence and was most common were maternal HIV status, maternal orphanhood, sex before HIV diagnosis/ART initiation, and sexual practices. The first two are strong indicators of vertical acquisition. While sexual activity is highly stigmatised, especially in populations with high HIV prevalence, a way to approach this topic may be to ask questions about sexual reproductive health knowledge and services available to them as a proxy to gauge an adolescent's sexual behaviour. In the absence of extensive personal information, acquiring these additional factors, when possible, can validate and improve the accuracy of the MOHA determinations made by the age cut-off.

This study does have its limitations. First, a logic tree is created based on a set of assumptions that may not hold for all cases. This limitation was mitigated by the inclusion of several assumptions for an iterative MOHA determination process. The results showed the average number of confirmatory branches for classifications were more than one, exhibiting high likelihood of accurate classification.

Second, some of the decision branches included are country specific and are influenced by the societal, political, economic, and historical context of the local HIV/AIDS epidemic. For example, the South African government's distrust in the origins of the virus and the efficacy of ART in post-apartheid South Africa led to the delayed rollout of ART nationally [75]. This also delayed the establishment of the proper infrastructure required for national rollout. However, these South African specific branches are lower in the logic tree and many of the factors used in earlier decision branches are relevant in other Sub-Saharan contexts. For example, other Sub-Saharan countries have similar trends to South Africa regarding experiencing sexual abuse at a young age, with Kenya, Nigeria, Uganda, and Zambia reporting incidence rates of about 20% for those aged 13 and younger. These rates increase by 10–20% for adolescents aged 14–15 [69]. Also, the belief in virgin cleansing and HIV witchcraft was also seen in Swaziland and Zimbabwe [76].

Another limitation is that neither participants nor their caregivers were asked directly what their most likely mode of HIV acquisition was, and thus, results were dependent on other self-

reported data. Some cohort participants only had data for one or two data collection points in the study, however, patient file data was included to compensate for some instances of missing data regarding age of ART initiation.

A strength of this study is the large sample size, combined with community-traced data collection methods ensuring the reach of those who may have been lost to care providing a representative sample, the patient file-verified data, and the robustness of the dataset, reaching 90.1% initiated adolescents within a municipality and covering a variety of social and clinical factors. The strength of the dataset allowed internal validity checks of branches used. The breadth of the data also allowed assessment of the strongest evidence for each MOHA determination—the exit branch number—as well as concentration of confirmatory evidence—the number of agreement branches for each adolescent.

The fact that the logic tree algorithm was tested in different samples–the cohort non-mothers, cohort mothers, and additional young mothers–demonstrates its reliability and responsiveness when applied to different demographic groups. The data used in the logic tree are not information that clinics have the capacity to collect. Thus, determining the sensitivity and specificity of age cut-offs provide a gauge for which cut-off to use depending on the interest of clinics and researchers. Lastly, this is the first study to compare different age cut-offs used for MOHA determination validated by adolescent life histories in an LMIC.

## Conclusion

Mode of HIV acquisition has been shown to differentiate the experience and care services needed by adolescents living with HIV. While all ALHIV should receive comprehensive, high quality HIV care, acknowledging and assessing MOHA in research and in care settings can play a vital role in understanding the distinct needs of ALHIV and designing suitable care to ensure treatment adherence. Incorporating MOHA in LMICs can also help create better differentiated service delivery, improving outcomes for adolescents living with HIV. This is important for reaching the UNAIDS 95-95-95 goal, as adolescent incidence rates are not decreasing as rapidly as other age groups in sub-Saharan Africa, and loss to follow-up remains high in this region.

## Acknowledgments

We thank all the adolescents who participated in this study, the respective field teams and support staff, and the University of Oxford and University of Cape Town for supporting this research.

## Author Contributions

**Conceptualization:** Eda He, Elona Toska.

**Data curation:** Siyanai Zhou.

**Formal analysis:** Eda He, Janke Tolmay, Siyanai Zhou.

**Methodology:** Eda He, Janke Tolmay, Siyanai Zhou, Wylene Saal.

**Supervision:** Elona Toska.

**Validation:** Janke Tolmay.

**Writing – original draft:** Eda He, Janke Tolmay.

**Writing – review & editing:** Eda He, Janke Tolmay, Siyanai Zhou, Wylene Saal, Elona Toska.

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
