## [Decision Letter · Decision Letter 0]

24 May 2022

PONE-D-22-11073Mode of HIV acquisition among adolescents living with HIV in resource-limited settings: a data-driven approach from South AfricaPLOS ONE

Dear Dr. Tolmay,

Thank you for submitting your manuscript to PLOS ONE. After careful consideration, we feel that it has merit but does not fully meet PLOS ONE’s publication criteria as it currently stands. Therefore, we invite you to submit a revised version of the manuscript that addresses the points raised during the review process.

We look forward to receiving your revised manuscript.

Kind regards,

Yogan Pillay, Phd

Academic Editor

PLOS ONE

Journal Requirements:

Additional Editor Comments:

This is an important area of study. However, there are significant challenges to the conceptualisation of the key question that the authors tried to answer (reviewer 1) and the lack of coherence and alignment as noted by reviewer 2). Given the importance of this issue for planning services as well as interventions the authors are encouraged to revise the manuscript taking into account the reactions of the reviewers.

Reviewers' comments:

Reviewer's Responses to Questions

**Comments to the Author**

1. Is the manuscript technically sound, and do the data support the conclusions?

Reviewer #1: Partly

Reviewer #2: Partly

2. Has the statistical analysis been performed appropriately and rigorously? 

Reviewer #1: Yes

Reviewer #2: I Don't Know

3. Have the authors made all data underlying the findings in their manuscript fully available?

Reviewer #1: No

Reviewer #2: No

4. Is the manuscript presented in an intelligible fashion and written in standard English?

Reviewer #1: No

Reviewer #2: No

5. Review Comments to the Author

Reviewer #1: The article has both strengths and weaknesses which are summarised below:

Strengths

1.Understanding mode of HIV acquisition (MOHA) has relevance for planning and provision of HIV services, and the study based on a large cohort of adolescents provides information that is relevant from a population and epidemiological perspective.

2.The study is methodologically sound (with some limitations as noted below).

Weaknesses

3.The article overstates the importance of mode of acquisition for individual patient care and support decisions. Such decisions should be based on individual patient need rather than on mode of acquisition. For example, whilst adolescents with vertically-acquired infection are at higher risk of experiencing cognitive delay and poor physical health, interventions aimed at mitigating the effects of these conditions, should be available to all adolescents, irrespective of MOHA.

4.The study’s aim and objective, findings and discussion/conclusions are not fully aligned, and the abstract and the article are likewise not fully congruent. For example, the article concludes with the statement that “incorporating MOHA in LMICs will also be necessary to reach the UNAIDS 90-90-90 goal as adolescent incidence rates are not decreasing as rapidly as other age groups and loss to follow-up remains high in these countries”. This is not supported by the content of the article, and a more modest conclusion based on the findings of this study would be more appropriate.

5.Some of the assumptions used in the study’s logic tree may not be justified, and may therefore limit its utility. In particular, it is problematic to assume that transmission is always sexually acquired, if the adolescent was sexually active before initiating ART. This should as a minimum be highlighted as a limitation.

6.Some of the statements about the health system do not make sense. For example, the statement that the study population included “adolescents who had ever initiated ART in a municipality in the Eastern Cape” is confusing, as there is no part of the Eastern Cape which is not in a municipality – or does it mean something else? It is not clear if the study includes only adolescents initiated in the public health sector, or whether (the presumably small number of) adolescents receiving care in the private sector are included.

Recommendation

Overall, I would recommend accepting the article providing that there is clearer alignment between and articulation of the aims, results, discussion and conclusion - both in the abstract and the text of the article. The article should focus on clearly reporting its finding related to methods for determining MOHA, and ensure that all conclusions are based on and fully justified by these findings.

Data privacy

Not all data are disclosed - restrictions seem appropriate.

Reviewer #2: Although the authors claim that there is a significant difference in the way that adolescents with HIV are treated dependent on whether their HIV was acquired from MTCT (vertical) or sexually acquired (horizontal), the case for this and the evidence was not compelling. It is not clear whether in resource-constrained environments of LMICs such as rural South Africa there can be different services. In the view of this reviewer for example adolescents with HIV need counselling around sexual and reproductive health and disclosure irrespective of how they acquired the HIV; adolescents need to be on ARTs for life and be virally suppressed irrespective of mode of acquisition.

So the case for the paper is not clearly articulated.

In addition the precise cut off age for determining the mode of age of HIV acquisition does not seem to be clearcut with some parameters increasing and others decreasing with age (viz sensistivity and specificity) and the area under the curve although statistically different probably does not mean much in real life.

Therefore the foundations for the paper and its conclusions are based on shaky grounds.

The writing of the paper is not clear and is unnecesarily complex and obtuse. The first paragraph in the discussion is used to illustrate "Emerging evidence has found variations in health determinants and needs among

adolescents living with HIV based on their mode of HIV acquisition, varying from familiarity

with the health system, available social support, and status awareness [2,7]. As this is a higher

risk group for loss to follow-up and HIV-related mortality, clinical services need to acknowledge

the differences in HIV experience between the two MOHAs and tailor services to these needs.

6. PLOS authors have the option to publish the peer review history of their article (what does this mean?). If published, this will include your full peer review and any attached files.

Reviewer #1: No

Reviewer #2: No

---

## [Author Response · Author response to Decision Letter 0]

11 Aug 2022

Reviewer 1

1. The article overstates the importance of mode of acquisition for individual patient care and support decisions. Such decisions should be based on individual patient need rather than on mode of acquisition. For example, whilst adolescents with vertically-acquired infection are at higher risk of experiencing cognitive delay and poor physical health, interventions aimed at mitigating the effects of these conditions, should be available to all adolescents, irrespective of MOHA. 

We wholeheartedly agree with the reviewer’s feedback and believe that all ALHIV should receive care that is tailored to their needs. We also agree that not endorsing the idea that MOHA should determine availability of treatments. However, in the low-resource, high-demand settings where the ALHIV in this study live, healthcare providers often have to make quick decisions on the focus of their care. We believe that having some understanding of the history of HIV acquisition of the young person they are providing care for can be valuable, in supporting healthcare providers think about possible needs of the ALHIV they are providing care for, such as disclosure, cognitive development, physical health and SRH needs. However, we agree that it is not – by itself – sufficient to shape HIV care services. We have edited the discussion to reflect this point:

“As ALHIV is a higher risk group for loss to follow-up and HIV-related mortality, research and clinical services should consider MOHA to understand differences in HIV pathology and needs to tailor services for better outcomes [19, 69-72]. Assessing MOHA can improve upon research on the effects of length of infection on long-term needs and outcomes of ALHIV in LMIC settings, as most of the current research on transition of care based on MOHA comes from high-income countries [7]. It can also help inform the provision of differentiated health services, such as support in transitioning from paediatric to adult care for adolescents with vertically acquired HIV, and sexual reproductive health and disclosure counselling for adolescents with sexual HIV acquisition”

And in the conclusion:

“While all ALHIV should receive comprehensive, high quality HIV care, acknowledging and assessing MOHA in research and in care settings can play a vital role in understanding the distinct needs of ALHIV and designing suitable care to ensure treatment adherence. Incorporating MOHA in LMICs can also help create better differentiated service delivery, improving outcomes for adolescents living with HIV.”

2. The study’s aim and objective, findings and discussion/conclusions are not fully aligned, and the abstract and the article are likewise not fully congruent. For example, the article concludes with the statement that “incorporating MOHA in LMICs will also be necessary to reach the UNAIDS 90-90-90 goal as adolescent incidence rates are not decreasing as rapidly as other age groups and loss to follow-up remains high in these countries”. This is not supported by the content of the article, and a more modest conclusion based on the findings of this study would be more appropriate. 

We thank and agree with the reviewer that the alignment of the aims and methodology in the manuscript can be adjusted to better fit the scope of the findings. 

While we believe that the impact of having differentiated service delivery and tailored care for ALHIV does indeed have broader consequences for decreasing rates of HIV infection, we have edited the conclusion text to better reflect the focus of the study as follows: 

“While all ALHIV should receive comprehensive, high quality HIV care, acknowledging and assessing MOHA in research and in care settings can play a vital role in understanding the distinct needs of ALHIV and designing suitable care to ensure treatment adherence. Incorporating MOHA in LMICs can also help create better differentiated service delivery, improving outcomes for adolescents living with HIV. This is important for reaching the UNAIDS 95-95-95 goal, as adolescent incidence rates are not decreasing as rapidly as other age groups in sub-Saharan Africa, and loss to follow-up remains high in this region.”

3. Some of the assumptions used in the study’s logic tree may not be justified, and may therefore limit its utility. In particular, it is problematic to assume that transmission is always sexually acquired, if the adolescent was sexually active before initiating ART. This should as a minimum be highlighted as a limitation. 

We thank the reviewer for this comment, and agree that the limitations of the logic tree, based on its assumptions, should be acknowledged. Each individual step of the logic tree has limitations, but this is why we use multiple assumptions, as well as calculating the total number of confirmatory factors for each participant who enters the logic tree. 

We have added the following to the discussion section as a study limitation: 

“This study does have its limitations. First, a logic tree is created based on a set of assumptions that may not hold for all cases. This limitation was mitigated by the inclusion of several assumptions for an iterative MOHA determination process. The results showed the average number of confirmatory branches for classifications were more than one, exhibiting high likelihood of accurate classification.”

4. Some of the statements about the health system do not make sense. For example, the statement that the study population included “adolescents who had ever initiated ART in a municipality in the Eastern Cape” is confusing, as there is no part of the Eastern Cape which is not in a municipality – or does it mean something else? It is not clear if the study includes only adolescents initiated in the public health sector, or whether (the presumably small number of) adolescents receiving care in the private sector are included. 

We thank the reviewer for highlighting this potentially confusing cohort description. ‘In a municipality’ refers to a specific municipality within the Eastern Cape Province. All of the adolescent participants receive care in the public health sector. 

To improve clarity, we have edited the text to read: 

“Adolescents who had ever initiated ART in the public health sector in the Buffalo City municipality of the Eastern Cape, South Africa, were community-traced to be invited to face-to-face interviews.”

5. Overall, I would recommend accepting the article providing that there is clearer alignment between and articulation of the aims, results, discussion and conclusion - both in the abstract and the text of the article. The article should focus on clearly reporting its finding related to methods for determining MOHA, and ensure that all conclusions are based on and fully justified by these findings. 

We thank the reviewer for highlighting important points regarding the alignment, aims and scope of the paper. The paper has been revised considerably to improve alignment and add more emphasis on not only the clinical application of the findings, but also the relevance for research methodology. For example, having an algorithm that can distinguish between potential recent and vertical HIV acquisition has enabled us to use this as a covariate in further analyses (presented at CROI 2021 and AIDS 2022), and may be helpful for other similar studies. 

Reviewer 2

1. Although the authors claim that there is a significant difference in the way that adolescents with HIV are treated dependent on whether their HIV was acquired from MTCT (vertical) or sexually acquired (horizontal), the case for this and the evidence was not compelling. It is not clear whether in resource-constrained environments of LMICs such as rural South Africa there can be different services. In the view of this reviewer for example adolescents with HIV need counselling around sexual and reproductive health and disclosure irrespective of how they acquired the HIV; adolescents need to be on ARTs for life and be virally suppressed irrespective of mode of acquisition. 

So, the case for the paper is not clearly articulated. 

Thank you for this comment – it was helpful that both reviewers highlighted the need to be clearer on the need and utility of knowing MOHA for ALHIV. We agree that comprehensive, high-quality care should be given to all ALHIV, regardless of MOHA. However, this paper aims to add another layer of information to care providers to support tailoring how they approach patients, rather than whether they do it. For example, if a healthcare provider is seeing a 15-year-old patient who started ART/was diagnosed before 10 years old, they can tailor their psychosocial support screening to the needs of an ALHIV with vertically-acquired HIV. 

Having a validated measure of MOHA can also be useful to researchers, to help understand the experiences and outcomes of adolescents living with HIV, based on the timing of their HIV acquisition. 

We have edited the discussion and conclusion to elaborate more on this: 

“As ALHIV is a higher risk group for loss to follow-up and HIV-related mortality, research and clinical services should consider MOHA to understand differences in HIV pathology and needs to tailor services for better outcomes [19, 69-72]. Assessing MOHA can improve upon research on the effects of length of infection on long-term needs and outcomes of ALHIV in LMIC settings, as most of the current research on transition of care based on MOHA comes from high-income countries [7]. It can also help inform the provision of differentiated health services, such as support in transitioning from paediatric to adult care for adolescents with vertically acquired HIV, and sexual reproductive health and disclosure counselling for adolescents with sexual HIV acquisition”

And in the conclusion:

“While all ALHIV should receive comprehensive, high quality HIV care, acknowledging and assessing MOHA in research and in care settings can play a vital role in understanding the distinct needs of ALHIV and designing suitable care to ensure treatment adherence. Incorporating MOHA in LMICs can also help create better differentiated service delivery, improving outcomes for adolescents living with HIV.”

2. Although there are clear statistical numbers, viz Sensitivity, specificity and Area under the curve, it is not clear that any one age chosen between 10 and 15 will have much more practical difference on the identification of the mode of acquisition. In addition, the precise cut off age for determining the mode of age of HIV acquisition does not seem to be clearcut with some parameters increasing and others decreasing with age (viz sensitivity and specificity) and the area under the curve although statistically different probably does not mean much in real life.

This paper aimed to ascertain an age cut-off for MOHA that has the highest accuracy, validated against a logic tree algorithm that provides more contextual information such as delayed diagnosis, high risk of sexual acquisition through early sexual debut, etc. Prior studies had used age 10 as a proxy cut-off, but reported concerns about overestimating the rate of ALHIV with sexually acquired HIV due to late progressors (REFS). However, we wanted to confirm that even when we take into account the complex lives of adolescents, we used the complex dataset to confirm these age cut-offs generally determined in biomedical datasets. 

Since we found that there is not a largely significant difference between the different age cut-offs, we continue to recommend the proxy already in use and that maximises sensitivity, which is 10 years old. 

Furthermore, when ascertaining the most suitable age cut-off, we chose to maximise sensitivity in our methodology, choosing age 10 for both the cohort and young mothers based on its high sensitivity. Since sexually infected adolescents were chosen as ‘true positives’ in the sensitivity/specificity analyses, we are maximising our screening for picking up adolescents in this group specifically. 

3. The authors have not taken into account that MTCT of HIV is decreasing with the successful implementation of PMTCT programmes throughout SSA. As a result successive cohorts of adolescents will have less and less vertical transmission with sexual transmission being the main driver. The discussion also does not make clear case for how the mode of transmission affects clinical care in practice. 

We argue that this shifting in dynamic would further strengthen the case for accurate identification of MOHA for researchers and health care providers alike, since, as mentioned in the previous response, our methodology is optimised to detect adolescents with sexually acquired HIV. 

However, while the scaling up of PMTCT services in South Africa has been successful in decreasing the rate of MTCT, data show that proportions of ALHIV with vertical acquisition remain significant. According to UNICEF data from 2020 (https://data.unicef.org/resources/hiv-estimates-for-children-dashboard/), the number of mother-to-child transmissions was on par with the numbers of sexually acquired HIV among adolescents in SSA. The most recent trend also shows a slightly faster rate of decline in sexually acquired adolescent HIV compared to MTCT. As seen in the Population-based HIV Impact Assessments (PHIA) biomarker data in SSA (Low et al., 2021), proportions of ALHIV with vertical acquisition remain high, with many unaware of their status. 

4. The writing of the paper is not clear and is unnecessarily complex and obtuse. The first paragraph in the discussion is used to illustrate "Emerging evidence has found variations in health determinants and needs among 

adolescents living with HIV based on their mode of HIV acquisition, varying from familiarity 

with the health system, available social support, and status awareness [2,7]. As this is a higher 

risk group for loss to follow-up and HIV-related mortality, clinical services need to acknowledge 

the differences in HIV experience between the two MOHAs and tailor services to these needs. 

We thank the reviewer for this feedback. The language use of the manuscript has been reviewed, and the writing in these and other examples have been simplified. We hope that this has made the manuscript more accessible. 

5. The scientific language is not always accurate. For example, in background it states: “The adolescent HIV incidence rate was 24% in 2018….”. 

Another example is the discussion is: “This also resulted in a delay in establishing the proper infrastructure to orchestrate such a feat” 

Again, the scientific language has been reviewed throughout the manuscript and edited for clarity and conciseness. 

The examples highlighted have been updated with the most recent statistics, and edited to read: 

“The adolescent HIV incidence rate was 29 cases per 100,000 in 2018 and projections indicate 1.8 million adolescents will be newly infected between 2018 and 2030 [11,15].” on page 3. 

And on page 20: 

“This also delayed the establishment of the proper infrastructure required for national rollout.”

---

## [Editor Report · Decision Letter 1]

8 Nov 2022

PONE-D-22-11073R1Mode of HIV acquisition among adolescents living with HIV in resource-limited settings: a data-driven approach from South AfricaPLOS ONE

Dear Dr. Tolmay,

Thank you for submitting your manuscript to PLOS ONE. After careful consideration, we feel that it has merit but does not fully meet PLOS ONE’s publication criteria as it currently stands. Therefore, we invite you to submit a revised version of the manuscript that addresses the points raised during the review process.

ACADEMIC EDITOR: The authors have adequately addressed the reviewer's comments. Upon review, there are additional minor edits that are needed to clarify the results and improve readability. Once addressed, the manuscript can be approved for publication.Background, first paragraph: sentence starting with ‘Further, facing the issues…’ appears to be incomplete.Background, second paragraph: provide reference for statement that the majority of current ALHIV is vertical transmission. Also, this seems to contradict the authors response to reviewers that in 2020 the numbers of vertically and sexually infected adolescents were about equal.Background, second paragraph: reference to most infected children pass away within their first year is outdated. This was true of untreated children early on in the epidemic but is no longer true as many vertically infected children are now living to adolescence and adulthood with ART. Sentence needs to be revised.Background, last paragraph: add a period after refs [18, 19, 5].Table 1: why does the title refer to N=1107 instead of N=1141?Table 1: mean/SD is calculated – confirm that data are normally distributed or change to median/IQR.MOHA allocation and confirmation, first paragraph: first sentence refers to 69.6% categorized as vertically infected with age cutoff of 10. This doesn’t match Table 2 which has 69.9% for mothers/non-mothers combined.MOHA allocation and confirmation, second paragraph: delete ‘the’ in first sentence (… we found that most adolescents were ….).MOHA allocation and confirmation, third paragraph: first sentence refers to 90.7% of participants with sexually acquired HIV using age cutoff of 10 (original allocation)This doesn’t match Table 2 (88.8%).MOHA allocation and confirmation, third paragraph: 4.6% vertically infected vs. 4.7% in the Table 2.Table 3: add in N’s for sexually acquired vs. vertically acquired.MOHA allocation and confirmation, fourth paragraph: this paragraph refers to the adolescents, but I think should refer to the additional young mothers. Please revise as needed.Table 4: add in N’s for sexually acquired vs. vertically acquired.Table 4: what does ‘Avg (SD) total agreement’ refer to?Determination of optimal cut-off age, first paragraph: delete duplicate ‘had’ in first sentence (… ART initiation at 10 years old had the highest sensitivity…).==============================

We look forward to receiving your revised manuscript.

Kind regards,

Catherine G. Sutcliffe

Academic Editor

PLOS ONE
---

## [Author Response · Author response to Decision Letter 1]

3 Jan 2023

3 January 2023

To: The Editor, PLOS ONE

Responses to Editor comments 

Note: We thank the editor for their comments and corrections on the manuscript. After careful revision, we have corrected the numbers for the cohort adolescents to reflect the correct sample size (n=1,107), and thus updated tables, text and figures throughout the manuscript where applicable. This can be seen in the track-changes version of the manuscript submitted.

1. Background, first paragraph: sentence starting with ‘Further, facing the issues…’ appears to be incomplete. 

Thank you for this comment, we have altered the sentence structure to include ‘while simultaneously’ instead of ‘further’, to improve clarity.

2. Background, second paragraph: provide reference for statement that the majority of current ALHIV is vertical transmission. Also, this seems to contradict the authors response to reviewers that in 2020 the numbers of vertically and sexually infected adolescents were about equal. 

This is a valid point, we have changed these sentences as indicated in point number 3. 

3. Background, second paragraph: reference to most infected children pass away within their first year is outdated. This was true of untreated children early on in the epidemic but is no longer true as many vertically infected children are now living to adolescence and adulthood with ART. Sentence needs to be revised.

We have revised the sentence to read:

“Although the risk of early mortality is increased for vertically infected children who are not initiated on ARV, research has shown that approximately 30% of vertically infected adolescents remain asymptomatic for 10 to 16 years.”

4. Background, last paragraph: add a period after refs [18, 19, 5].

Thank you, a period has been added. 

5. Table 1: why does the title refer to N=1107 instead of N=1141?

Thank you for this observation – the authors have addressed this inconsistency when updating the cohort sample size throughout the analysis. 

6. Table 1: mean/SD is calculated – confirm that data are normally distributed or change to median/IQR.

The numbers have been changed to median and IQR for better accuracy.

7. MOHA allocation and confirmation, first paragraph: first sentence refers to 69.6% categorized as vertically infected with age cutoff of 10. This doesn’t match Table 2 which has 69.9% for mothers/non-mothers combined.

This error has been corrected in the revised sample size. 

8. MOHA allocation and confirmation, second paragraph: delete ‘the’ in first sentence (… we found that most adolescents were ….).

Thank you for bringing this to our attention, the ‘the’ has been removed from the sentence. 

9. MOHA allocation and confirmation, third paragraph: first sentence refers to 90.7% of participants with sexually acquired HIV using age cutoff of 10 (original allocation). This doesn’t match Table 2 (88.8%).

The paragraph data was correct, and the table has been updated to reflect the correct numbers. 

10. MOHA allocation and confirmation, third paragraph: 4.6% vertically infected vs. 4.7% in the Table 2.

This was a rounding inconsistency. The table is correct, so the paragraph was edited to read ‘4.7%’. 

11. Table 3: add in N’s for sexually acquired vs. vertically acquired.

N’s for totals have been added. 

12. MOHA allocation and confirmation, fourth paragraph: this paragraph refers to the adolescents, but I think should refer to the additional young mothers. Please revise as needed.

Appreciation for picking up this unclarity. The text was edited here to clarify that this is referring to the additional young mothers, by replacing ‘adolescents’ with ‘young mothers’.

13. Table 4: add in N’s for sexually acquired vs. vertically acquired.

 N’s for each branch have been added.

14. Table 4: what does ‘Avg (SD) total agreement’ refer to?

This refers to the average number of affirmative answers to each branch after filtering into the vertical or sexual tree. In other words, on average, how many of the logic tree questions did each participant answer ‘yes’ to. For clarity, the table heading has been changed to ‘Avg affirmative answers’. 

15. Determination of optimal cut-off age, first paragraph: delete duplicate ‘had’ in first sentence (… ART initiation at 10 years old had the highest sensitivity…). 

Thank you, the duplicate ‘had’ has been deleted. 

Responses to additional comments: 3 January 2023

2. In the Methods section please revise the informed consent statement to reflect whether written/verbal informed consent was obtained from all participants for inclusion in the study.

On Page 4 paragraph 2, the word “written” was added to clarify the type of informed consent obtained. 

“Voluntary written informed consent was also obtained from these additional participants” was added to Page 5 paragraph 1 (for the additional young mothers).

---

## [Editor Report · Decision Letter 2]

20 Jan 2023

Mode of HIV acquisition among adolescents living with HIV in resource-limited settings: a data-driven approach from South Africa

PONE-D-22-11073R2

Dear Dr. Tolmay,

We’re pleased to inform you that your manuscript has been judged scientifically suitable for publication and will be formally accepted for publication once it meets all outstanding technical requirements.

Kind regards,

Catherine G. Sutcliffe

Academic Editor

PLOS ONE

Additional Editor Comments (optional):

In the last sentence of the 'MOHA allocation and confirmation' paragraph, please confirm that the % listed for the non-mothers (20.8%) is correct (should 19.0% be reported instead for the algorithm allocation).
---

## [Editor Report · Acceptance letter]

15 Feb 2023

PONE-D-22-11073R2 

Mode of HIV acquisition among adolescents living with HIV in resource-limited settings: a data-driven approach from South Africa 

Dear Dr. Tolmay:

I'm pleased to inform you that your manuscript has been deemed suitable for publication in PLOS ONE. Congratulations! Your manuscript is now with our production department. 

Kind regards, 

on behalf of

Dr. Catherine G. Sutcliffe 

Academic Editor

PLOS ONE